# Real-World Evaluation of Immune-Related Endocrinopathies in Metastatic NSCLC Patients Treated with ICIs in Romania

**DOI:** 10.3390/cancers17071198

**Published:** 2025-03-31

**Authors:** Simona Coniac, Mariana Cristina Costache-Outas, Ionuţ-Lucian Antone-Iordache, Ana-Maria Barbu, Victor Teodor Bardan, Andreea Zamfir, Andreea-Iuliana Ionescu, Corin Badiu

**Affiliations:** 1Department of Medical Oncology, Hospice Hope Bucharest, 023642 Bucharest, Romania; simona.horlescu@drd.umfcd.ro; 2Department of Endocrinology, “Carol Davila” University of Medicine and Pharmacy, 020021 Bucharest, Romania; anamaria.lg.barbu@stud.umfcd.ro (A.-M.B.); teodor.gi.bardan@stud.umfcd.ro (V.T.B.); andreea.zamfir@stud.umfcd.ro (A.Z.); corin.badiu@umfcd.ro (C.B.); 3Department of Endocrinology, Coltea Clinical Hospital, 030167 Bucharest, Romania; costache.mariana@gmail.com; 4Department of Radiotherapy, Coltea Clinical Hospital, 030167 Bucharest, Romania; antoneiordachelucian@stud.umfcd.ro; 5Department of Medical Oncology, Colțea Clinical Hospital, 030167 Bucharest, Romania; 6C.I. Parhon National Institute of Endocrinology, 011863 Bucharest, Romania

**Keywords:** metastatic non-small-cell lung cancer, immune check-point inhibitors, endocrine immune-related adverse events, predictive biomarkers, Romania

## Abstract

High-quality real-world data may offer indirect evidence of the impact of immunotherapy in challenging, under-represented metastatic lung cancer patients and areas of unmet need in daily clinical practice. Innovative gold standard immune check-point inhibitors (ICIs) provide increased overall survival in these patients with the cost of immune-related adverse events (irAEs). This research aimed to identify if autoimmune endocrinopathies might be predictive for ICI responses and to distinguish certain medical situations with impacts on ICI efficacy. The actual Romanian retrospective study evidenced a statistically significant correlation between endocrine irAEs and overall survival in metastatic non-small-cell lung cancer patients treated with ICIs and classified several co-medications with negative impacts on responses to immunotherapy. These real-world data inform the medical community about disease presentation, predictive factors, treatment effectiveness, and survival in complex clinical circumstances.

## 1. Introduction

Despite revolutionary and constantly updated therapies, lung cancer remains the first mortality cause in cancer patients worldwide, including Romania [1,2]. Immune check-point inhibitors (ICIs) are the state-of-the-art standard treatment in first and second lines [3,4], refining overall survival (OS) in metastatic non-small-cell lung cancer (mNSCLC) without oncogenic addiction [5,6,7,8,9,10,11]. However, international guidelines mostly support evidence from randomized clinical trials (RCTs), which do not always match real-world cancer patient populations [12]. Real-world data (RWD) studies are needed to generate high-quality real-world evidence (RWE) for subpopulations under-represented in RCTs [13].

Immune-targeted therapies break the cancer immune tolerance, restoring T-cell recognition against tumor cells and restraining the cytotoxic T-lymphocyte-associated antigen 4 (CTLA-4) and the programmed cell death 1 (PD-1)/ligand 1 (PD-L1) pathways. Even if ICIs accomplish a durable clinical response and predictive biomarkers direct the efficacy trend, the majority of patients show resistance to immunotherapy [14]. Besides innate or acquired resistance to ICIs, environmental factors such as microbiota, diet, and co-medications are associated with poorer clinical outcomes [15,16].

Immune-related endocrinopathies are frequent and common adverse events during ICI treatment, as reported in RCTs and RWE [5,6,7,8,9,10,11,17,18]. Endocrine immune-related adverse events (E-irAEs) emerged as a class of possible predictive and prognostic biomarkers for ICI response and OS, though the literature data provide conflicting results [19,20,21]. Following our previous research [22,23], our RWD retrospective study thoroughly analyzed mNSCLC patients treated with ICIs in a tertiary-level hospital in Romania, exploring new multiple clinical parameters and the correlation between them and endocrine irAEs using a multidisciplinary team (MDT) approach.

## 2. Materials and Methods

### 2.1. Study Population

This retrospective, observational, non-interventional cohort analysis utilized data of lung cancer patients treated with ICIs in a tertiary-level hospital, Coltea Clinical Hospital, Bucharest, Romania. The evaluation period started with the first approval of Nivolumab, an anti-PD-1 inhibitor, on 1 November 2017 in Romania until 31 July 2024, comprising a period of almost 7 years. Inclusion criteria identified adult metastatic non-small-cell lung cancer patients without an oncogenic driver mutation or after completed targeted therapy for a specific mutation who signed and agreed to ICI treatment according to Romanian National Protocols. Exclusion criteria comprised patients with the following at baseline: renal or liver failure, untreated oncogenic driver mutations, active Hepatitis B or Hepatitis C infection, active Human immunodeficiency virus (HIV) infection, active autoimmune diseases, active immunosuppressive treatment, pregnancy, or lactation. During the retrospective analysis, the frequencies of hospital visits monitored the specific ICI cycle once every two weeks (q2w), or every 3 weeks (q3w), or as needed. For endocrine irAE assessment, all mNSCLC patients were evaluated for thyroid functional tests (TFTs) at baseline and before every ICI cycle, with at least one follow-up value.

Ethical Institutional Review Board approval of the study was obtained before the study was conducted. We assessed lung cancer patients following European Society of Medical Oncology (ESMO) Guidance for Reporting Oncology Real-World evidence (GROW) criteria [13]. Baseline demographic and clinical characteristics of mNSCLC patients, type of used ICI, response to treatment, co-medication during ICIs, and overall survival were estimated using registered data. The imaging ICI’s efficacy followed response evaluation criteria in solid tumors (RECIST) v1.1 [24]. Endocrine irAEs were defined as any occurring autoimmune endocrinopathy during ICIs and were diagnosed, graded, treated, and followed by an endocrinologist in an MDT, following international guidelines [25,26]. For grading endocrine irAEs, we referred to Common Terminology Criteria for Adverse Events (CTCAE) v5.0 [27].

The main endpoint of the research was to investigate immune-mediated endocrinopathies as predictive events for ICI response and overall survival (OS). The secondary endpoints consisted in identifying clinical conditions with potential impacts on ICI efficacy.

### 2.2. Statistical Methods

For statistical analyses, JASP Version 0.19.1 and R version 4.3.2 were used [28]. The former was accessed through RStudio version 2024.12.0.467 [29].

Categorical variables were presented as number of patients and frequencies (percentages), while continuous variables were presented as mean and standard deviation.

We performed univariate logistic regressions to assess if the selected predictors influenced the appearance of irAEs or only endocrine irAEs, respectively. Odds ratios were calculated. Significant predictors were then included in two multivariate logistic regression models, one for each type of irAE. In order to investigate if the model fits were good, McFadden R² was calculated (a value between 0.2 and 0.4 is considered a good model fit).

Survival analysis was performed at first for the whole population and then only for the population treated with ICIs as first-line therapy. Several variables were tested: dyslipidemia, PD-L1 (<1, 1–49, ≥50), histological type, Diabetes Mellitus, surgery before ICIs, steroids during ICIs, proton-pump inhibitor (PPI) used during ICIs, opioids during ICIs, infections treated during ICIs, response to ICIs, all irAEs, endocrine irAEs, and adjuvant radiotherapy/chemotherapy; only significant results after applying the log-rank test were displayed, and restricted mean survival time (RMST) was calculated, and Kaplan–Meier plots were generated.

Predictors where survival curves did not cross were selected for univariate Cox proportional hazards regression. To check for the proportional hazard’s assumption, the cox.zph() function was used in R, and Schoenfeld residuals were calculated. Only variables where this assumption was met were displayed and then included in a multivariate Cox regression. Hazard ratios were calculated.

We did not adjust for multiple comparisons in order to control the probability of type II error. A *p*-value of <0.05 was considered statistically significant.

## 3. Results

Of 487 cancer patients registered and treated with ICIs over a period of almost 7 years in Coltea Clinical Hospital, Bucharest, Romania, we included 237 lung cancer patients in the study, as our baseline cohort. Other cancer site patients treated with immunotherapy were excluded, such as three cases of cancer of unknown primary, three of breast cancer, five of hepatocellular carcinoma, 15 of urothelial cancer, 18 of renal cell cancer (RCC), 51 of malignant melanoma (MM), and 155 of head and neck squamous cancer patients (HNSCC). We selected only lung cancer patients for homogeneity purposes and to avoid bias. As other cancer sites are treated with cervical surgery and radiotherapy, such as HNSCC patients, or combine ICIs with targeted therapies, such as RCC, or provide an aggressive natural history, for example MM, we applied these exclusion criteria for accurate evaluation of autoimmune endocrinopathies.

### 3.1. Study Population Analysis

#### 3.1.1. Lung Cancer Population at Baseline

Of 237 lung cancer patients treated with ICIs in our retrospective study, we classified 15 cases of small-cell lung cancer (SCLC) patients and 222 NSCLC patients. A total of 215 NSCLC cancer patients were treated for metastatic disease with first- or second-line ICIs. Seven NSCLC patients were treated with Durvalumab for stage III disease and consolidation after chemo-radiotherapy. For more appropriate statistical analysis, we excluded SCLC and stage III Durvalumab-treated patients, and the final cohort of the study found 215 metastatic NSCLC patients.

The flowchart of the study cohorts and exclusion criteria are described in Figure 1.

#### 3.1.2. Metastatic NSCLC Population Evaluated for Endocrine irAEs

The research focused on 215 cases of mNSCLC patients, who were evaluated from different perspectives. Patient demographic and clinical characteristics assessed at baseline (meaning the initiation of ICI therapy) included age, sex, body mass index (BMI), dyslipidemia (any type) and Diabetes Mellitus, Eastern Cooperative Oncology Group (ECOG) performance status, histological features, number and sites of metastasis at baseline, presence of EGFR mutations or ALK rearrangements, and PD-L1 status. All these data are shown in Table 1. Treatment characteristics, such as type of ICI used in the first and second line, number of ICI cycles, response to ICIs, type of therapy before ICIs, and radiotherapy for metastatic sites, are detailed in Table 2.

This cohort counted 148 male and 67 female adults, with a median age of 64 years old. One hundred and sixty-nine (78.6%) patients had a good performance status, with an ECOG between 0 and 2. Pathology features discovered 146 adenocarcinoma mNSCLC patients and 69 squamous, respectively. Eighty-six (40%) patients had just one metastatic site and almost 29% had more than three sites before ICI therapy. Ten mNSCLC patients had been treated with targeted therapy before ICIs for Epidermal Growth Factor Receptor (EGFR) mutations, and one patient had already received therapy for Anaplastic Lymphoma Kinase (ALK) rearrangements. Even if all stage IV NSCLC cases (squamous and non-squamous) are recommended for programmed death-ligand 1 (PD-L1) immunohistochemistry (IHC) testing, the test was not assessed in 29 patients. PD-L1 expression was acknowledged from IHC as TPS (tumor proportion score, meaning a report between PD-L1 positive tumor cells and the sum of PD-L1 positive and negative tumor cells). Almost 70% of patients were treated in the first line, mostly with Pembrolizumab, and Nivolumab was the main ICI in the second line, comprising 54 patients. The average time on the ICIs was 8.9 months and 38.1 weeks, respectively. ICI’s efficacy was evaluated by RECIST criteria and showed 112 patients with a partial response, defined as partial response, stable disease, or clinical benefit, or a combination of these. Before ICI therapy, 41 patients were treated by surgery, and 46 patients received radio-chemotherapy or systemic treatment for an earlier stage. Radiotherapy (RT) for brain metastasis, stereotactic or whole brain, occurred in 46 patients.

### 3.2. Analysis of Clinical and Co-Medication Variables During ICIs

This retrospective study cohort revealed mNSCLC patients in different clinical situations. The average time from the first diagnostic gesture (surgery or biopsy) till the first therapeutic intervention, such as surgery, chemotherapy, radiotherapy, or immunotherapy, was 3.7 months. Thirty-eight mNSCLC patients experienced at least one more neoplasia during their lifetime, actual cancer being the second primary for 19 patients and synchronous with other types of cancer sites for 19 patients. During ICI therapy, we noticed registered data of co-medications, such as steroids, antibiotics, antifungal and retroviral treatment, opioids, and proton-pump inhibitors (PPIs). Of 74 patients treated for infectious diseases, 53 received antibiotics alone or antifungals alone or combinations, 2 patients postponed ICIs for active tuberculosis treatment, and 24 patients took retroviral medication for Hepatitis B (nine patients), Hepatitis C (six cases), SARS-COV2 infection (six cases), two cases of Varicella zoster virus, and one case of Human immunodeficiency virus (HIV) infection. More than half of the patients were treated with steroids for different purposes, such as co-medication in chemotherapy protocols or for symptomatic brain or bone metastases, as specified in Table 3. One hundred and thirty-one patients were treated with PPIs. Seventy-two (33.5%) patients had a symptomatic pain, controlled by minor opioids in 37 patients or major opioids in 35 cases. All mNSCLC patients were evaluated for TFTs at baseline and before each cycle of immunotherapy or at least one follow-up value. If patients reported modified TFTs or had clinical or biological suspicion of endocrine dysfunction, endocrinologist specialist consultation was required. Additional functional tests to support autoimmune endocrinopathy diagnostics involved thorough evaluation of other hormones, alongside clinical symptoms. Low cortisol and low adrenocorticotropic hormone (ACTH) were useful to diagnose hypophysitis and low cortisol, and increased ACTH and abnormal ACTH stimulation tests (Synacthene test) were detected in primary adrenal insufficiency (PAI), respectively. Besides modified TFTs, the presence of thyroid peroxidase (TPO) antibodies was used to assess thyroiditis. Consistent increased glycemia and glycated hemoglobin needed mandatory specialist evaluation and diabetes mellitus treatment. Fifty-five mNSCLC patients experienced immune-related adverse events (irAEs) during immunotherapy. Clinical features and co-medications of mNSCLC patients during immunotherapy are presented in Table 3.

### 3.3. Analysis of Endocrine Immune-Related Adverse Events

Immune-related adverse events were classified as endocrine irAEs and other irAEs. Since five patients developed both endocrine and non-endocrine irAEs and four patients experienced more than one affected gland, a total number of 55 patients experienced 65 irAEs during the study. Two patients treated with Nivolumab experienced hepatitis and myositis in one case and hepatitis and hypophysitis in another case. Three patients treated with Pembrolizumab experienced a combination of irAEs, such as hepatitis, PAI, and hypothyroidism in one case, nephritis and hypothyroidism in the other case, and colitis and PAI in the last case. The average time till endocrine irAE appearance was 6.8 months. In mNSCLC patients, 47 (21.8%) were diagnosed, treated, and followed for 51 endocrine irAEs, of which just one was classified as moderate to severe (Grade ≥ 3), due to primary adrenal insufficiency (PAI). Hypothyroidism was identified in 18 cases that needed hormone replacement treatment. Our retrospective cohort discovered 10 cases of hypophysitis, 5 cases of PAI, and 3 cases of type I Diabetes Mellitus (DM). Thirteen mNSCLC patients developed 14 other irAEs: five cases of hepatitis, three cases of colitis, two cases of myositis, and one case of myocarditis, as well as one nephritis, one dermatitis, and one anemia. Seven other-irAEs were classified as mild (Grade ≤ 2), and seven cases as moderate to severe (Grade ≥ 3). All other-irAEs were treated with methylprednisolone. All endocrine and other-irAEs are detailed in Table 4.

mNSCLC patients that developed irAEs during the study showed a 15.7 months average time on immunotherapy. Thirty-seven (67.2%) of patients who experienced irAEs were treated with Pembrolizumab and fourteen (25.4%) with Nivolumab, respectively. Most endocrine irAEs occurred during Pembrolizumab therapy, with 58.2% of patients, followed by Nivolumab with 21.8% of cases. All endocrine and other immune-related adverse events during ICIs are detailed in Table 5.

### 3.4. Statistical Analysis

When testing predictors that influence the appearance of every type of irAE (Table 6), we saw that surgery before ICIs, the number of weeks on ICIs, and response to ICIs increased the appearance of irAEs, while administering PPIs during ICIs decreases the appearance of irAEs. When we introduced these predictors in a multivariate logistic regression, only the response to ICIs was statistically significant (Table 7), as it accounted for most of the effect.

We applied the same statistical model for endocrine irAEs (Table 8) and a PD-L1 higher than 50%; the number of weeks on ICIs and response to ICIs all significantly increased the odds of developing endocrine irAEs when analyzed with a simple logistic regression model. After we introduced these predictors in a multiple regression model (Table 9), the response to ICIs was the only significant variable, probably accounting again for most of the effect. The McFadden R^2^ was closer to the 0.2 value, which indicates a good fit.

Statistically analyzed predictors for survival identified several variables for increased overall survival, both in the entire cohort population and in first-line ICI treated populations, such as surgery before ICI therapy, good response to ICIs, and any irAEs or endocrine irAEs, respectively. Kaplan–Meier curves for endocrine irAEs in these two analyzed cohorts are exemplified in Figure 2 and Figure 3. Negative predictors for survival in both populations were co-medication, such as steroids, and PPIs. Besides these biomarkers, in the first-line ICI cohort composed of 150 mNSCLC patients, infections treated during ICIs had a negative impact on survival, while positive PD-L1 status increased survival. Forty-six mNSCLC patients with just one or two cycles of ICIs were not included in this analysis, as they were not alive at sixty months. All predictors are presented in Table 10 and Table 11, with individual *p* values.

Univariate and multivariate Cox regression for selected predictors concerning the entire cohort of 215 mNSCLC patients and the first-line ICI-treated 150 mNSCLC patients during the study are thoroughly presented in Table 12 and Table 13.

When accounting for the whole study population, the positive impact of surgery before ICI and negative impact of steroids and PPIs as co-medication were preserved and statistically significant when using a univariate model. Introducing these variables in a multiple Cox regression showed that the predictor “PPIs during ICI” lost its significance, as PPIs were frequently co-administered with steroids.

In the subgroup with first-line ICIs, infections treated during ICI, a PD-L1 value < 1%, steroids, and PPI medication increased the risk of dying, while a PD-L1 value > 50% and surgery before ICI decreased it. When we entered these predictors in a multiple Cox regression model, the only variables with a significant negative outcome were “Infections treated during ICI” and “PD-L1 < 1%”, while suffering a surgery before the treatment with ICIs showed a significant decrease in the risk of dying.

## 4. Discussion

Several first-line combinations of platinum-based chemotherapy (ChT) plus an immune-checkpoint inhibitor successfully confirmed improved overall survival compared with ChT alone for mNSCLC patients, regardless of tumor PD-L1 status. In metastatic non-squamous and squamous non-small-cell carcinoma, RCTs for Pembrolizumab [30,31], Atezolizumab with or without bevacizumab (non-squamous only) [32,33], Cemiplimab [34], and Nivolumab/Ipilimumab [35] plus ChT unequivocally proved efficacy, and indications of use were approved by the Food and Drug Administration (FDA) and European Medicine Agency (EMA) [36,37,38,39,40]. A Durvalumab–Tremelimumab–ChT regimen is approved only by the FDA [41]. First-line monotherapy ICIs in tumor PD-L1 ≥ 50% is the standard of care according to clinical trials [42,43,44]. Second-line therapy for previously ChT-treated, PD-L1 inhibitor-naïve mNSCLCs, irrespective of tumor PD-L1 expression, is monotherapy ICIs with Nivolumab [45,46], Atezolizumab [47], and Pembrolizumab (only in PD-L1 ≥ 1%) [48], as the OS benefit was irrefutably evidenced.

In Romania, non-oncogene mNSCLC patients are treated following European Society for Medical Oncology guidelines [4], but not all of ICI indications are reimbursed [49]. The vast majority of patients is treated with first-line Pembrolizumab and second-line nivolumab, as our retrospective study evidenced. There is small to absent clinical experience with the 9LA regimen (first-line Nivolumab plus Ipilimumab combined with two cycles of chemotherapy) or Cemiplimab in mNSCLC (recently approved in December 2024), and this situation may have an impact on dealing with special patient populations, treating rare or late-onset immune-related adverse events. Even if ICI clinical trials [30,31,32,33,34,35] included only mNSCLCs in good performance status (PS), such as ECOG PS ≤ 1, and international guidelines recommend immunotherapy to ECOG PS ≤ 2 patients [3,4], real-life medical practice in oncology may be a different challenge. In our retrospective study, we identified 21.4% of ECOG PS = 3 mNSCLC patients treated with just one or two cycles of immunotherapy. For a more homogeneous cohort and clearer statistical analysis, these patients should have been excluded from the study. Bearing in mind the small sample size of mNSCLC patients, and to preserve the challenges and limitations of real-practice in Romania, where salvage therapy still used, we kept these patients in our retrospective research. Almost 60% of the entire cohort presented more than two sites of metastatic disease before ICIs and brain or liver metastasis were present in 20% of patients. This reality is completely different from the patients included in the RCTs, RWD being critical in understanding how RCTs are translated into clinical practice. On the other hand, really good RWE is needed because, as Pellat published in 2023 [50], only 8% of RWE sources were from more than one country; half of the studies were conducted in Asia, and 87% of the studies were retrospective. From our knowledge, this observational study on mNSCLC patients is the second one to report RWE from Romania, an Eastern European country. Furthermore, the mean time from diagnosis till the first therapeutic gesture was 3.7 months during the analyzed period, meaning a possible immortal time bias. Forty percent of mNSCLC patients were heavily pretreated before ICIs with surgery or radio-chemotherapy, and 38.6% of patients received palliative RT for brain or bone metastases or to control loco-regional disease. On top of the high metastatic burden of the mNSCLC patients’ cohort study, thirty-eight patients (17.6%) presented synchronous or metachronous neoplasia, as specified above, challenging further therapies for already profoundly pretreated patients or requiring alternative treatments for other neoplasia. In a nutshell, our entire cohort of mNSCLC patients was part of a significant share, unlike mNSCLC patients included in RCTs.

Clinical characteristics at baseline acknowledged 53% of mNSCLC patients diagnosed with dyslipidemia and 20% with diabetes mellitus. Moreover, 46.5% of patients were overweight. Registered data for co-medication evaluated 74 patients (34.4%) with treated infections during ICIs, such as antibiotic, antifungal, and retroviral treatments, 120 patients (55.8%) treated with steroids, 131 patients (60.9%) with PPIs, and 72 patients (33.5%) with minor or major opioids. As already widely ascertained, the gut microbiota’s impressive diversity reflects dietary habits, environmental exposure, antibiotic usage, and lifestyle factors [51,52]. The human microbiota dynamically interacts with the host’s immune system, implying an essential role in immune maturation, maintenance of mucosal barriers, and regulation of inflammatory responses [53,54]. Chalabi M. previously published in 2020 the negative impact of cumulative use of multiple or prolonged antibiotics on treatment outcomes of immunotherapy [55]. A similar retrospective study evaluating antibiotic use in advanced cancer patients treated with immunotherapy highlighted clear worse survival outcomes [16]. In conclusion, the intestinal microbiome emerged as a factor that might modulate ICI efficacy and may be altered by antibiotic use [56]. Similarly, long-term usage of proton-pump inhibitors (PPIs) before ICI therapy seems to have a dismal impact on overall survival [57] because microbial alterations in gut microbiota are more significant than with antibiotics or other drug use [58]. Chen B et al. reported in 2022 a meta-analysis and systematic review that confirmed PPIs’ alteration to the gut microbiome and influence on response to ICIs [59]. In our research as well, PPI usage and treated infections during ICI therapy demonstrated a harmful effect on overall ICI outcome. Moreover, our retrospective study lines up with the literature data in proving the detrimental effect of steroid use on ICI efficacy [15]. Considering that ICIs are designed to enhance the immune system’s inherent antitumor activity, and steroids have immunosuppressive properties, inevitably the association might have undesirable consequences. Available data suggest that 10 mg of prednisone, as substitution therapy, has no influence over ICI outcomes, and this dose is often used in treating hypophysitis, as Cortisone Acetate has limited availability in Romania. If in other irAEs the steroid or immunosuppressant’s need could have detrimental effects on the antitumor efficacy and this effect could be dose and time-exposure related, in endocrine irAEs the substitutive therapy is not expected to counteract or impact the antitumor ICI activity. Larger doses of steroids used before or soon after ICI initiation were correlated with poorer clinical outcomes [15]. Our cohort study identified palliative usage of steroids and PPIs for brain metastasis, bone, or brain radiotherapy, meaning patients with severe disease during ICIs. Also, steroids and PPIs were used in chemoprevention emesis protocols. To specifically assess the cause–effect relationship between steroid and PPI usage and ICI efficacy, dedicated specific protocol studies should be implemented. Our cohort study was a retrospective, non-interventional in nature, with no pre-specified protocol. Consequently, we could not assess the causality effect of steroids and PPIs used in lung cancer patients treated with immunotherapy, and it was not a main point of the study. To finalize the assessment of co-medication used in our cohort, it is worth revealing opioid usage during ICI treatment. Alongside the central nervous system, opioid receptors are also expressed in the gastrointestinal tract, especially on neurons of the enteric nervous system, intestinal epithelial cells, and immune cells, indicating a possible effect on gut barrier function [60,61]. Therefore, opioid usage might lead to dysregulated immune response and dysbiosis [62,63,64], and, not surprisingly, inferior outcomes in cancer patients treated with ICIs [65]. To summarize, co-medication is crucial during ICIs in mNSCLC patients in need of steroids and PPIs alongside palliative bone or whole brain radiotherapy or chemotherapy emesis prevention or requiring opioids for pain management. Not only co-medication influences gut microbiota, but there is an evolving hypothesis of interactions between microbiota and the endocrine system [66].

The incidence of endocrinopathies in the whole population has dramatically changed over the past decades, influenced by lifestyle factors. Golden S.H. et al. reported a comprehensive review including articles and cohorts of patients from the United States, indicating at least a 5% prevalence of endocrine disorders. The most prevalent disorders were diabetes mellitus and metabolic dysfunctions, with no published population-based data on hypothyroidism incidence [67]. During immunotherapy in cancer patients, the incidence of endocrine disorders as immune-related adverse events was completely different, reported as high and increasing over years of getting more experience using ICIs. If in 2017 Barroso-Sousa R. et al. reported a 3.8% incidence of all grade hypothyroidism with anti-CTLA-4 antibodies, 7% with anti-PD-1, and 13.2% with a combination [68], in 2023, Won S.Y. et al. published an increased incidence of immune-related hypothyroidism of 4.4% with CTLA-4 inhibitors, 13.7% with PD-1 inhibitors, and a decreased incidence of 9.7% with an anti-CTLA-4/PD-L1 antibody combination [69]. To summarize, the incidence of immune-mediated endocrine disorders was increasing as more cancer patients and additional indications of usage in further diverse cancer sites were legally approved by regulatory authorities. Taking into consideration that patients excluded from RCTs, such as patients with active brain metastasis, active infections, steroid usage above 10 mg of daily prednisone or equivalent, were not investigated for endocrine irAEs, RWD were even more crucial to provide experience from clinical practice in such challenging populations of cancer patients.

The actual retrospective study reported endocrine irAEs that followed the incidence, severity grading, and well-known management from RCTs [5,6,7,8,9,10,11]. Immune-related thyroid disorder is well established to be the most regular and recurrent autoimmune endocrinopathy in the course of immune check-point inhibitors, mainly associated with anti-PD-1 monotherapy [8,9,43,70]. Clinical presentation is challenging, as new onset hypothyroidism or a transient hyperthyroid state followed by an euthyroid state or hypothyroidism might occur. ICI protocols recommend TFTs each cycle to diagnose thyroid toxicity early, and oncology practice adheres to close surveillance. Also, in our cohort study, we evaluated TFTs each cycle and monitored thyroid function as developing toward immune-related adverse events or recovering to euthyroid status after subclinical alteration. Even if TFTs were normal each ICI cycle, we assessed thyroid function regularly, as guideline protocols recommended. We reported patients with transitional modified TFTs to better acknowledge the possibility of temporary effects of ICIs on thyroid function. Our retrospective analysis did not convey any case of Graves’ disease or thyroid storm or myxedema coma, as the literature rarely reports such [71,72,73,74]. Thyroid immune-related adverse events revealed an exciting research endpoint, associated with better overall survival in some studies [19,22,75,76] and conflicting results in others [20,21,77]. ICI-induced hypophysitis is a thought-provoking clinical situation, because non-specific symptoms or adrenal crisis, a potential life-threatening emergency may occur in cancer patients treated with immunotherapy. When immune-related hypophysitis is suspected, the diagnose needs detection of low morning cortisol and low ACTH values. If thyroid gland is also affected, low TSH and low FT4 values are present. If gonadotroph lines are affected, low gonadal hormones plus low luteinizing hormone (LH) and follicle stimulating hormone (FSH) are also present. Additionally, electrolytes should be obtained. Posterior pituitary involvement may present with hyponatremia. Ir-hypophysitis is more related to and associated with improved response to anti-CTLA4 antibodies [68,78]. Primary adrenal insufficiency (PAI) stands also as a sporadic immune-related endocrinopathy that mimics symptoms of cancer progression, and a high suspicion should be well maintained during treatment to avoid adrenal crisis [79,80]. If low morning cortisol levels and high ACTH values are found inconclusive, a cosyntropin (Synacthene) test could be performed [81]. For patients already on substitution therapy with prednisone, cortisol testing was performed after a 24 h pause of substitution therapy plus a cosyntropin test for reassurance. Laboratory testing often reveals hyponatremia, hyperkalemia, and hypoglycemia. Unlike secondary adrenal insufficiency, mineralocorticoid deficiency might be caused by PAI, and plasma renin and aldosterone levels could be useful for differential diagnosis [82]. Our research identified just one moderate to severe endocrine irAE, and that was one case of PAI, associated with a Nivolumab/Ipilimumab combination and treated with Prednisone hormonal therapy (the only available therapy). Diabetes mellitus as a worsening preexisting DM or new onset of insulin-dependent DM is one of the most infrequent autoimmune endocrinopathies during ICI therapy in mNSCLC patients, with an incidence below 0.6%, as reported in RCTs [9,11,30,32,33,34,43]. To help differentiate type 1 from type 2 DM, low levels of insulin and C peptide might be useful [83,84]. Despite the usual management of treatment with methylprednisolone in irAEs, immune-related-DM does not benefit from steroid therapy [84]. In our study, type I DM occurred in three mNSCLC patients treated with ICIs, and insulin therapy was initiated by a specialist.

The main endpoint of the study was to investigate the possible relationship between endocrine irAEs and immunotherapy response, and our study fulfilled this imperative outcome. The overall survival was statistically significant and improved in mNSCLC patients that developed endocrine irAEs compared to the entire cohort of the study (*p*-value = 0.002) and compared to first-line ICI-treated patients (*p*-value = 0.004). Nevertheless, the literature acknowledges conflicting results. On the one hand, there are data that prove better overall survival for cancer patients treated with immune check-point inhibitors [19,22,85,86], and, conversely, inconsistent statistics that diminish the value of irAEs [20,21]. Enlightenment could arise from late-onset and long-lasting irAEs that are underreported but common events during ICI therapy. In our cohort research, we discovered an interesting and challenging clinical practice of mNSCLC patients treated in the first line with Pembrolizumab (plus ChT for the first 4–6 cycles, in PD-L1 < 50%, and according to the histological report) beyond 35 cycles, as studied in RCTs [8,9,30,31]. Taking into consideration that this first-line cohort of patients treated with Pembrolizumab represented almost half of the entire cohort study, we could assess the incidence and prevalence of late-onset and long-lasting endocrine irAEs, as specified before. Additionally, this first-line cohort of patients received steroids and PPIs as co-medication for chemotherapy emesis prevention. On the one hand, longer exposure of responding mNSCLC patients to ICIs produced a higher incidence of endocrine irAEs; the question raised was how to differentiate between steroid-induced endocrinopathy and immune-related endocrine toxicity? On the other hand, endocrine irAEs are not treated with steroids as are all other irAEs, so the detrimental effect of steroids did not counteract this kind of toxicity. Finally, to better distinguish and identify the causality effect of co-medication and late-onset and long-lasting endocrine irAEs in responding to ICI mNSCLC patients, specific protocol studies should be implemented. Consequently, patients have an assurance of survival up to the onset of that toxicity, better explained by the immortal-time bias. Specific endocrine irAEs, such as PAI or hypophysitis or DM, usually appear after several months of immunotherapy or even years, and the short period of follow-up or incomplete reporting of duration and resolution of toxicities definitely underrates the clinical magnitude of endocrine irAEs. RWE should be a mandatory consecutive support to RCTs, as it delivers data on a broader spectrum of real-life cancer patients on long time scales.

In addition, our study revealed surgery before ICIs as another predictor, with a positive impact on lung cancer patients’ survival. Deeply investigating the medical trajectory of our cohort, surgery was identified as curative gesture in early stages of the disease, with a more favorable prognostic factor, allowing the patient a long time of progression free survival until relapse. Surgical removal of the primary lung tumor, if resectable, continues to stand as the cornerstone of lung cancer treatment [87,88]. Surgery itself does not impact ICI survival; it just grants alongside immunotherapy an extended survival.

In conclusion, this retrospective cohort research of mNSCLC patients investigated the possible correlation between immune-mediated endocrinopathies as predictive events for ICI response and overall survival, as a core endpoint outcome. The secondary endpoint of the study fulfilled its purpose by identifying clinical conditions with potential impacts on ICI efficacy. As thoroughly discussed above, the analysis acknowledged significant statistical correlations between endocrine irAEs and ICI responses and proved the negative impact of specific co-medications during ICIs, such as steroids, PPIs, and opioids. Nevertheless, being a retrospective cohort from real-life clinical practice, the patient population was different from RCTs, more heterogenous, frailer, and with more severe disease. Even if the sample size appeared to be rather small, thinking of one tertiary-level hospital in Romania, not the high-volume Lung Cancer Institute, the number of patients could be matched with RCTs that are multicentered, multinational, and delivered in specialized oncologic medical institutions [43,89]. A pooled analysis of KEYNOTE-021, -189 and -407 studies that investigated Pembrolizumab plus platinum-based chemotherapy for mNSCLC patients with stable brain metastasis showed a proportion of patients with brain metastasis of 13.17% [90]. In addition, all patients were in good performance status, with an ECOG 0-1. In contrast, our study included 21.4% of patients with brain metastasis and patients in moderate performance status, with an ECOG 0-2, and 21.4% of patients with an ECOG-3, normally excluded from ICI RCTs. As RCTs were conducted in homogenous cancer populations, such as histopathological subtypes, positive and negative PD-L1 cohorts, ICI in monotherapy, or plus ChT, the current study included all treated mNSCLC patients in a specific period of time, leading to a more heterogenous patient population. To divide this population into specific subtypes would have increased homogeneity but prevented a fluent statistical analysis.

We assert the limitations of our retrospective research. Nonetheless, this study cohort was a transparent and honest piece of real-life data, unveiling the challenges, imperfections and limitations that oncologists faced in daily practice. Our research has several limitations to be listed. First of all, our retrospective-in-nature observational study did not a have an a priori-defined protocol and used registered data, resulting in potential selection bias of patients. Secondly, the retrospective eligibility criteria of mNSCLC treated with ICIs shaped a heterogeneous cohort of patients and pointed toward confusion bias. Multiple and varied types of treatment during the study might have induced an immortal-time bias. Finally, the frequency and methods of tumor assessment were not always standardized or recorded; some patients were lost to follow-up after one or two cycles of ICIs, implying missing data and information bias. Although RWD studies are more expected to present all these biases compared with clinical trials, RWD research might generate evidence from sub-populations under-represented in RCTs, building higher generalizability than clinical trials. Nonetheless, this study cohort was a transparent and honest piece of real-life data, unveiling the challenges, imperfections, and limitations that oncologists face in daily practice.

## 5. Conclusions and Future Directions

Good quality RWE is needed for optimizing and improving access to innovative treatments in clinical practice. Our research stated clear evidence and statistically significant proof of close correlations between variable clinical factors and ICI outcomes in mNSCLC patients. Endocrine irAEs could be considered predictive biomarkers for response to immunotherapy, but steroids and opioids used as co-medication for mNSCLC patients can have a dismal effect on efficacy to immune check-point inhibitors and must be further investigated in prospective studies. Collaborative efforts towards nationwide and international standardized collection of data sources must be encouraged to boost the relevance and future quality of publications and their potential impact on oncology practice. Treating cancer patients might be challenging, but teamwork and dialogue in a multidisciplinary team (MDT) approach can help physicians to improve a patient’s quality of life and boost the efficacy of cancer therapies.

## Figures and Tables

**Figure 1 cancers-17-01198-f001:**
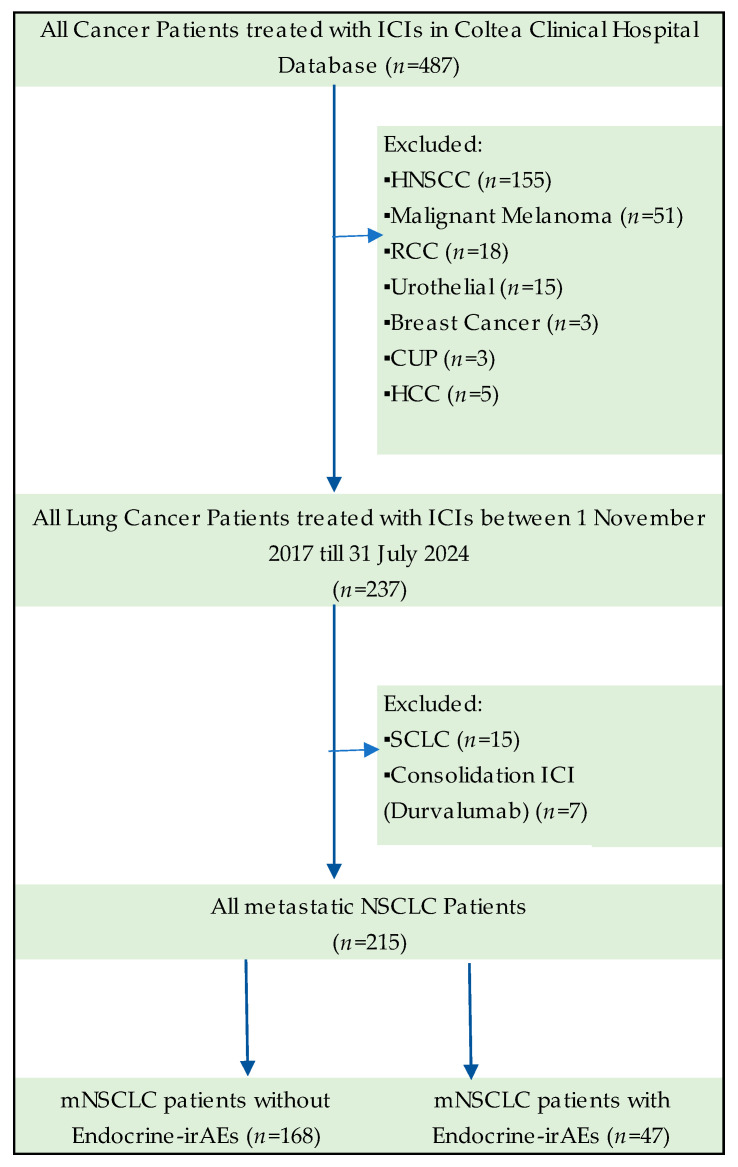
Flowchart of the study cohorts.

**Figure 2 cancers-17-01198-f002:**
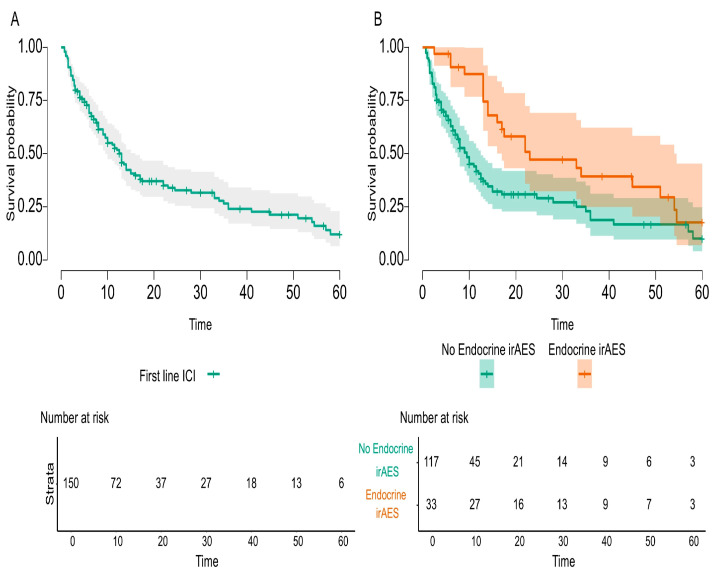
Overall survival for first-line ICI-treated mNSCLC patients with endocrine irAEs compared to all first-line ICI-treated mNSCLC patients during the study.

**Figure 3 cancers-17-01198-f003:**
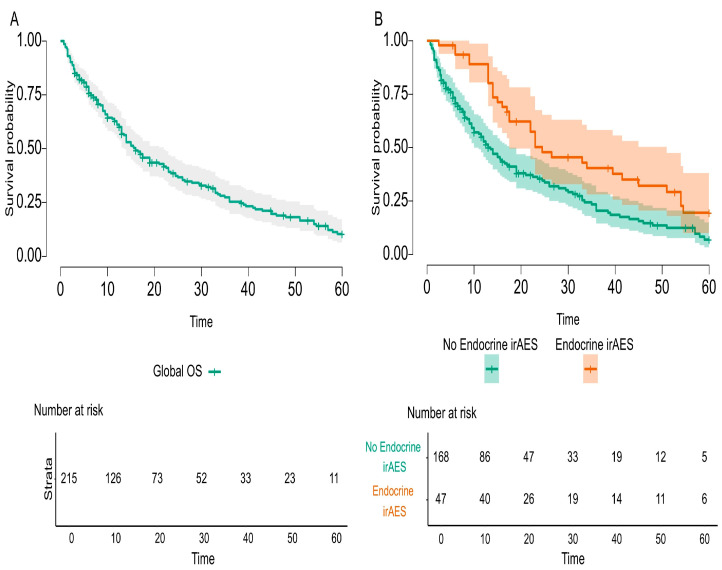
Overall Survival for mNSCLC patients with endocrine irAEs compared to global population during the study.

**Table 1 cancers-17-01198-t001:** Patient demographic and clinical characteristics of mNSCLC patients treated with ICIs, assessed at baseline (before ICI initiation).

Characteristic	*n* = 215
Age	
Median (range)—years	64 (34–82)
>65 years	118 (54.9)
<65 years	97 (45.1)
Male sex—no. (%)	148 (68.8)
Smoker—no. (%)	
Yes	110 (51.1)
No	14 (6.5)
Not known	91 (42.3)
Body Mass Index (kg/m^2^)—no. (%)	
<18.5	22 (10.2)
>18.5 <24.9	93 (43.2)
>24.9	100 (46.5)
Dyslipidemia	
Yes	114 (53.0)
No	101 (47.0)
Diabetes Mellitus	
Yes	44 (20.4)
No	171 (79.5)
ECOG performance status	
0–2	169 (78.6)
3	46 (21.4)
Histologic features—no. (%)	
Adenocarcinoma	146 (68.2)
Squamous cell carcinoma	69 (31.8)
Number of metastatic sites before ICI—no. (%)	
1 site	86 (40.0)
2 sites	67 (31.1)
3 sites	34 (15.8)
≥4 sites	28 (13.0)
Metastases sites before ICI—no. (%)	
Pulmonary +/− Pleural	143 (66.5)
Bone	68 (31.6)
Lymph nodes	55 (25.6)
Brain	46 (21.4)
Liver	44 (20.5)
Adrenal	40 (18.6)
Other (spleen, skin)	14 (6.5)
EGFR-mutated NSCLC—no. (%)	10 (4.6)
ALK rearrangements—no. (%)	1 (0.4)
PD-L1 Tumor Proportion Score—no. (%)	
<1%	71 (33.0)
1–49%	46 (21.4)
>50%	69 (32.0)
Not assessed	29 (13.5)

**Table 2 cancers-17-01198-t002:** Treatment characteristics of mNSCLC patients treated with ICIs, evaluated for endocrine irAEs.

Immunotherapy Type and Treatment Details	*n* = 215
ICI used and line of treatment—no. (%)	
First line	150 (69.8)
Pembrolizumab/ChT	101 (46.9)
Pembrolizumab Monotherapy	38 (17.6)
Nivolumab/Ipilimumab + ChT	11 (5.1)
Second line	65 (30.2)
Nivolumab	54 (25.1)
Atezolizumab	5 (2.3)
Pembrolizumab	6 (2.8)
Average time on ICI (range)—months	8.9 (0.5–60.9)
Number of ICI cycles	
>1≤2	46 (21.4)
≥3	169 (78.6)
ICI Treatment Response ^1^—no. (%)	
Complete Response	1 (0.46)
Partial Response *	112 (52.0)
Progressive Disease	37 (17.2)
Not Assessed	65 (30.2)
Treatment before first line ICI ^2^—no. (%)	87 (40.4)
Surgery	41 (19.0)
RT-ChT/ChT	46 (21.4)
Palliative Radiotherapy ^3^—no. (%)	83 (38.6)
Tumor and mediastinal lymph nodes	22 (10.2)
Brain metastasis	46 (21.4)
Bone metastasis	30 (13.9)
Average time on Total Treatment ^4^ (range)—months	19.6 (0.7–87.0)

^1^ Patients were evaluated by RECIST v1.1. * Partial response was assessed as partial response, stable disease, clinical benefit, or a combination of these. ^2^ Chemotherapy at physician’s choice, as docetaxel, gemcitabine, navelbine, paclitaxel, pemetrexed. ^3^ Patients were treated with RT for one or more sites. ^4^ Total Treatment is composed by all curative and palliative treatments during the study. ChT, chemotherapy; ICI, immune check-point inhibitor.

**Table 3 cancers-17-01198-t003:** Clinical Features of mNSCLC patients during immunotherapy.

Characteristic	*n* = 215
Synchronous/metachronous neoplasia	
Yes	38 (17.6)
No	177 (82.3)
Treated infections during ICIs	
Yes	74 (34.4)
No	141 (65.6)
Steroids during ICIs	
Yes	120 (55.8)
Brain metastasis	35 (16.2)
Radiotherapy for brain or bone metastasis	57 (26.5)
Chemoprevention emesis therapy	99 (46.0)
No	95 (44.2)
PPIs during ICIs	
Yes	131 (60.9)
No	84 (39.0)
Opioids during ICIs	
Yes	72 (33.5)
No	143 (66.5)
Immune-related adverse events ^1^	55 (25.6)
Endocrine irAEs	47 (21.8)
Other irAEs	13 (6.0)
Average time from diagnosis till first treatment (range)—months	3.7 (0.0–96)

^1^ Five patients developed both endocrine and non-endocrine irAEs. The first neoplasia was represented by head and neck cancer (H&N) in six cases, NSCLC in three cases, one anal, one ovarian, one prostate, two cases of malignant melanoma (MM), two cases of breast cancer, three cases of cervical cancer, one case of cancer of unknown primary, and one case of hematologic disease, chronic lymphatic leukemia (CLL). For metachronous disease, we identified five cases of H&N cancer, three cases of colorectal cancer (CRC), three cases of CLL, one ovarian cancer, one non-Hodgkin’s Lymphoma, one MM, one renal cell carcinoma, two cases of urothelial cancer, one hepatocellular cancer and one case of NSCLC. One female patient experienced four neoplasia types during the study: synchronous CRC and NSCLC and metachronous breast and cervical cancer.

**Table 4 cancers-17-01198-t004:** Summary of irAEs during ICIs for metastatic NSCLC patients.

Characteristic	*n* = 215	irAE Grade ≤ 2	irAE Grade ≥ 3	Treatment
Average time till E-irAEs (range)—months	6.8 (0.0–49)			
**Endocrine irAEs ^1,2^**	**47 (21.8)**	**50 (23.2)**	**1 (0.46)**	
Thyroiditis	33 (15.3)	33 (15.3)	0	Levothyroxine as needed
Transient modified TFTs	10 (4.6)	10 (4.6)	0	None
Hypothyroidism ^3^	18 (8.3)	18 (8.3)	0	Levothyroxine
Hyperthyroidism	5 (2.3)	5 (2.3)	0	None
Hypophysitis ^4^	10 (4.6)	10 (4.6)	0	Levothyroxine/Prednisone *
Primary Adrenal Insufficiency	5 (2.3)	4 (1.8)	1 (0.46)	Prednisone
Diabetes Mellitus	3 (1.4)	3 (1.4)	0	Insulin Therapy
**Other irAEs ^5^**	**13 (6.0)**	**7 (3.2)**	**7 (3.2)**	
Hepatitis	5 (2.3)	3 (1.4)	2 (0.9)	Methylprednisolone
Colitis	3 (1.4)	2 (0.9)	1 (0.46)	Loperamid/Methylprednisolone
Myocarditis	1 (0.46)	1 (0.46)	0	Methylprednisolone
Myositis	2 (0.9)	0	2 (0.9)	Methylprednisolone
Nephritis	1 (0.46)	1 (0.46)	0	Methylprednisolone
Dermatosis	1 (0.46)	0	1 (0.46)	Topic/Methylprednisolone
Anemia	1 (0.46)	0	1 (0.46)	Methylprednisolone

^1^ Five patients developed both endocrine and non-endocrine irAEs. ^2^ Four patients experienced more than one affected gland. ^3^ Patients who had subclinical hypothyroidism did not need therapy. ^4^ One patient experienced hypophysitis involving thyroid, adrenal, and gonadotropic deficiencies and needed hormonal replacement therapy for all three glands. ^5^ One patient experienced both hepatitis and myositis irAEs. *In Romania, Prednisone is used for substitution therapy.

**Table 5 cancers-17-01198-t005:** Summary of irAEs during ICIs for metastatic NSCLC patients—continued.

Characteristic	*n* = 55	irAEs Grade ≤ 2	irAEs Grade ≥ 3
Average time on ICI for mNSCLC EirAE patients (range)—months	15.7 (2.1–60.9)		
**ICI used in all irAE patients**	**55**	**57**	**8**
Atezolizumab	1 (1.8)	1 (1.8)	0
Nivolumab *	14 (25.4)	14 (25.4)	2 (3.6)
Pembrolizumab ª	37 (67.2)	40 (72.7)	3 (5.45)
Nivolumab / Ipilimumab	3 (5.45)	2 (3.6)	3 (5.45)
**ICI used in endocrine irAE patients ^1,2^**	**47 (85.4)**	**50**	**1**
Atezolizumab	1 (1.8)	1 (1.8)	0
Nivolumab *	12 (21.8)	12 (21.8)	0
Pembrolizumab ª	32 (58.2)	35 (63.6)	0
Nivolumab/Ipilimumab °	2 (3.6)	2 (3.6)	1 (1.8)
**Thyroiditis ^3^**	**33 (60.0)**	**33 (60.0)**	**0**
Atezolizumab	1 (1.8)	1 (1.8)	0
Nivolumab	8 (14.5)	8 (14.5)	0
Pembrolizumab	23 (41.8)	23 (41.8)	0
Nivolumab/Ipilimumab °	1 (1.8)	1 (1.8)	0
**Hypophysitis ^4^**	**10 (18.1)**	**10 (18.1)**	**0**
Nivolumab	2 (3.6)	2 (3.6)	0
Pembrolizumab	8 (14.5)	8 (14.5)	0
**Primary Adrenal Insufficiency**	**5 (9.0)**	**4 (7.2)**	**1 (1.8)**
Nivolumab	1 (1.8)	1 (1.8)	0
Pembrolizumab	2 (3.6)	2 (3.6)	0
Nivolumab/Ipilimumab °	2 (3.6)	1 (1.8)	1 (1.8)
**Diabetes Mellitus**	**3 (5.45)**	**3 (5.45)**	**0**
Pembrolizumab	2 (3.6)	2 (3.6)	0
Nivolumab	1 (1.8)	1 (1.8)	0
**ICI used in Other irAE patients**	**13 (23.6)**	**7 (12.7)**	**7 (12.7)**
Nivolumab *	3 (5.45)	2 (3.6)	2 (3.6)
Pembrolizumab	8 (14.5)	5 (9.0)	3 (5.45)
Nivolumab/Ipilimumab	2 (3.6)	0	2 (3.6)

^1^ Five patients developed both endocrine and non-endocrine irAEs. ^2^ Four patients experienced more than one affected gland. ^3^ Patients who had subclinical hypothyroidism did not need therapy. ^4^ One patient experienced hypophysitis involving thyroid, adrenal, and gonadotropic deficiencies and needed hormonal replacement therapy for all three glands. * Two patients treated with Nivolumab experienced a combination of irAEs, such as both hepatitis and myositis in one case and both hepatitis and hypophysitis in another case. ª Three patients treated with Pembrolizumab experienced a combination of irAEs, such as hepatitis, PAI, and hypothyroidism in one case, nephritis and hypothyroidism in one case, and colitis and PAI in another case, respectively. Three patients treated with Pembrolizumab experienced more than one affected gland. ° One patient treated with combination Nivolumab/Ipilimumab experienced PAI and hypothyroidism.

**Table 6 cancers-17-01198-t006:** Univariate logistic regression for all irAEs.

Variable		Odds Ratio (95% CI)	*p*-Value
BMI (range)		1.042 (0.986 to 1.101)	0.143
Age < 65 (yes)		1.509 (0.816 to 2.792)	0.19
Sex (Male)		1.016 (0.524 to 1.971)	0.962
Smoking (yes)		1.200 (0.649 to 2.219)	0.561
Preexisting diabetes (yes)		0.962 (0.448 to 2.065)	0.921
Histological type (SC)		0.929 (0.480 to 1.799)	0.827
PD-L1	PD-L1 < 1% (yes)	0.607 (0.293 to 1.259)	0.180
	PD-L1 > 50% (yes)	1.788 (0.900 to 3.551)	0.097
Adjuvant RT/CHT (yes)		1.074 (0.509 to 2.264)	0.851
Surgery before ICI (yes)		2.215 (1.077 to 4.559)	0.031
First-line ICI (yes)		1.076 (0.550 to 2.107)	0.831
Time from diagnostic until start of treatment (months)		1.044 (0.992 to 1.100)	0.098
Weeks of ICI		1.009 (1.004 to 1.015)	<0.001
Infections treated during ICI (yes)		1.247 (0.660 to 2.355)	0.496
Steroids during ICI (yes)		0.695 (0.376 to 1.285)	0.246
PPIs during ICI (yes)		0.519 (0.279 to 0.966)	0.038
Opioids during ICI (yes)		0.678 (0.345 to 1.332)	0.259
Synchronous/metachronous tumor (yes)		1.438 (0.669 to 3.092)	0.352
Response to ICI (yes)		4.607 (2.260 to 9.390)	<0.001

ICI, Immune check-point inhibitor; irAEs, immune-related adverse events; PD-L1, programmed cell death ligand 1; PPI, proton-pump inhibitor; BMI, body mass index.

**Table 7 cancers-17-01198-t007:** Multivariate logistic regression for all irAEs.

Variable	Odds Ratio	*p*-Value	McFadden R^2^
Whole model		<0.001	0.109
Surgery before ICI	2.117 (0.930 to 4.818)	0.074	
Weeks of ICI	1.001 (0.994 to 1.008)	0.747	
PPIs during ICI (yes)	0.653 (0.325 to 1.310)	0.230	
Response to ICI (yes)	4.094 (1.820 to 9.210)	<0.001	

**Table 8 cancers-17-01198-t008:** Univariate logistic regression for endocrine irAEs.

Variable		Odds Ratio	*p*-Value
BMI		1.044 (0.986 to 1.101)	0.143
Age < 65 (yes)		1.691 (0.882 to 3.245)	0.114
Sex (male)		1.239 (0.605 to 2.536)	0.558
Smoking (yes)		1.383 (0.720 to 2.655)	0.331
Preexisting diabetes (yes)		1.065 (0.482 to 2.355)	0.876
Histological type (SC)		0.990 (0.495 to 1.980)	0.976
PD-L1	PD-L1 < 1% (yes)	0.451 (0.199 to 1.023)	0.057
	PD-L1 > 50% (yes)	2.090 (1.008 to 4.332)	0.047
Adjuvant RT/CHT (yes)		1.204 (0.556 to 2.609)	0.637
Surgery before ICI (yes)		1.643 (0.762 to 3.543)	0.205
First-line ICI (yes)		1.027 (0.507 to 2.082)	0.940
Time from diagnostic until start of treatment (months)		1.060 (0.996 to 1.128)	0.069
Weeks of ICI		1.011 (1.005 to 1.017)	<0.001
Infections treated during ICI (yes)		1.104 (0.562 to 2.168)	0.775
Steroids during ICI (yes)		0.701 (0.367 to 1.342)	0.284
PPIs during ICI (yes)		0.532 (0.277 to 1.023)	0.059
Opioids during ICI (yes)		0.617 (0.298 to 1.278)	0.194
Synchronous/metachronous tumor (yes)		1.135 (0.495 to 2.602)	0.764
Response to ICI (yes)		9.111 (3.669 to 22.625)	<0.001

ICI, immune check-point inhibitor; irAEs, immune-related adverse events; PD-L1, programmed cell death ligand 1; PPI, proton-pump inhibitor; BMI, body mass index.

**Table 9 cancers-17-01198-t009:** Multivariate logistic regression for endocrine irAEs.

Variable	Odds Ratio	*p*-Value	McFadden R^2^
Whole model		<0.001	0.17
PD-L1 > 50% (yes)	1.769 (0.782 to 4.003)	0.171	
Weeks of ICI	1.003 (0.997 to 1.010)	0.343	
Response to ICI (yes)	8.686 (2.734 to 27.597)	<0.001	

**Table 10 cancers-17-01198-t010:** Sixty-months survival table for the entire cohort during the study.

	Number	Survival Rate (%)	RMST	Log-Rank Test *p*-Value
Global	N = 215	10.4%	23.592	-
PD-L1 < 1%	No N = 115	11.1%	25.391	0.018
	Yes N = 71	8.4%	17.710	
Surgery before ICI	No N = 174	4.4%	19.861	<0.001
	Yes N = 41	29.7%	37.251	
Steroids during ICI	No N = 95	16.4%	30.647	<0.001
	Yes N = 120	4.3%	17.254	
PPIs during ICI	No N = 84	15.1%	30.448	<0.001
	Yes N = 131	6.8%	18.591	
Response to ICI	No N = 102	1.5%	13.025	<0.001
	Yes N = 113	19.8%	32.815	
Any type of irAEs	No N = 160	7.4%	20.981	0.004
	Yes N = 55	16.8%	30.613	
Endocrine irAEs	No N = 168	6.9%	20.955	0.002
	Yes N = 47	19.5%	32.220	

CI, immune check-point inhibitor; irAEs, immune-related adverse events; PD-L1, programmed cell death ligand 1; PPI, proton-pump inhibitor.

**Table 11 cancers-17-01198-t011:** Sixty-months survival table for mNSCLC patients treated with first-line ICIs.

	Number	Survival Rate (%)	RMST	Log-Rank Test *p*-Value
Global	N = 150	12.1%	22.032	-
PD-L1 < 1%	No N = 92	15.1%	26.007	0.003
	Yes N = 55	6.8%	15.707	
PD-L1 > 50%	No N = 88	6.5%	18.472	0.029
	Yes N = 59	19.6%	27.493	
Surgery before ICI	No N = 119	4.3%	17.279	<0.001
	Yes N = 31	29.7%	36.577	
Steroids during ICI	No N = 43	23%	33.057	<0.001
	Yes N = 107	6.1%	16.816	
PPIs during ICI	No N = 39	23.9%	34.069	<0.001
	Yes N = 111	6.1%	16.808	
Infections treated during ICI	No N = 96	16.5%	25.217	0.036
	Yes N = 54	6.5%	16.975	
Response to ICI	No N = 73	9.2%	8.779	<0.001
	Yes N = 77	21.3%	33.245	
Any type of irAEs	No N = 111	10.7%	19.274	0.008
	Yes N = 39	15.6%	29.780	
Endocrine irAEs	No N = 117	10%	19.033	0.004
	Yes N = 33	17.7%	31.832	

ICI, immune check-point inhibitor; irAEs, immune-related adverse events; PD-L1, programmed cell death ligand 1; PPI, proton-pump inhibitor; BMI, body mass index.

**Table 12 cancers-17-01198-t012:** Univariate and multivariate Cox regression for selected predictors concerning the entire cohort during the study.

Variable	Hazard Ratio	*p*-Value
Dyslipidemia (yes)	0.7737 (0.5658 to 1.058)	0.108
Surgery before ICI (yes)	0.3722 (0.24 to 0.5773)	<0.001
Steroids during ICI (yes)	2.139 (1.543 to 2.963)	<0.001
PPIs during ICI (yes)	1.89 (1.362 to 2.621)	<0.001
Multivariate model		
Dyslipidemia (yes)	0.7720 (0.5631 to 1.0583)	0.107
Surgery before ICI (yes)	0.3608 (0.2316 to 0.5619)	<0.001
Steroids during ICI (yes)	2.1876 (1.1463 to 4.1748)	0.018
PPIs during ICI (yes)	1.0379 (0.5401 to 1.9945)	0.91

**Table 13 cancers-17-01198-t013:** Univariate and multivariate Cox regression for selected predictors concerning mNSCLC patients treated with first-line ICIs during the study.

Variable	Hazard Ratio	*p*-Value
Infections treated during ICI (yes)	1.519 (1.023 to 2.253)	0.038
PD-L1 < 1% (yes)	1.812 (1.211 to 2.711)	0.004
PD-L1 > 50% (yes)	0.6342 (0.42 to 0.9576)	0.03
Steroids during ICI (yes)	2.306 (1.434 to 3.708)	<0.001
Surgery before ICI (yes)	0.3538 (0.208 to 0.6019	<0.001
PPI during ICI (yes)	2.437 (1.489 to 3.987)	<0.001
Multivariate model		
Infections treated during ICI (yes)	1.6036 (1.0705 to 2.4024)	0.022
PD-L1 < 1% (yes)	1.9025 (1.1076 to 3.2679)	0.020
PD-L1 > 50% (yes)	1.2668 (0.6988 to 2.2964)	0.436
Steroids during ICI (yes)	1.7727 (0.4248 to 7.3972)	0.432
Surgery before ICI (yes)	0.3085 (0.1768 to 0.5383)	<0.001
PPI during ICI (yes)	1.3090 (0.2834 to 6.0447)	0.73

ICI, immune check-point inhibitor; irAEs, immune-related adverse events; PD-L1, programmed cell death ligand 1; PPI, proton-pump inhibitor; BMI, body mass index.

## Data Availability

The raw data supporting the conclusions of this article will be made available by the authors without undue reservation.

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
