# Peer review of "Real-World Evaluation of Immune-Related Endocrinopathies in Metastatic NSCLC Patients Treated with ICIs in Romania"

_cancers, 2025, doi:10.3390/cancers17071198_

Round 1

Reviewer 1 Report

Comments and Suggestions for Authors

The authors stated that RWD studies are necessary to generate high-quality RWE for subpopulations under-represented in RCTs, and I agree.
I am asking the authors to provide   the incidence of   endocrinopathies in the whole   population and in patients differing from those of RCTs.   For instance: WHO PS 3, pretreatment, brain met, active infections, concomitant tumors, steroid use > 10 mg daily of prednisone or equivalent, etc. In table 1 for instance the steroid use is not included.

Line 199: “Half of them were smokers and for 48.8% of the patients smoking was not toxic exposure” could you clarify please?

Lines 229-230    How cortisol and ACTH evaluation was performed in patients already on steroids?

Table 3: Transitory hyperthyroidism is usually a short window of subsequent hypothyroidism, sometimes captured, sometimes not, in my opinion it should not be calculated except when it persists, and that event is quite rare from literature and from my experience. Moreover, how did you include the transient modification of TFTs?

Table 4: clarify better the patient’s number and the events number, otherwise it is not clear and immediate.

Table 5: treatment duration and treatment response indicate that patients without progression remained on therapy longer and obviously could develop more AES, in table 6 only response to ICI remained significant and this is confirmation. Moreover, the positive effect of surgery could be related to a clinical more favorable disease situation allowing these patients to be operated on and not to the surgery itself.

Table 7: PDL1≥50% effect is probably related to a better response, can you confirm that?

In table 9: the significative effect of irAES on survival has been often reported, and related to multiple factors, among them the longer exposure of responding patients to treatment which increases the incidence of irAEs and the underlining common mechanisms of tumor response and irAES. Moreover, with endocrine irAES we do not have the detrimental effect of the steroids needed to counteract the toxicity.

363: indicate 9LA regimen as combo immune (NIVO/IPI)

Lines 368-370: do you refer to all ECOG PS 3 patients of the table 1 or 23 % of them who received only one-two cycles? If all patients with PS3 received only 1-2 cycles, their exclusion from the eligibility is justified, in my opinion.

Line411: Hypophysitis is usually treated with substitutive therapy with Cortone Acetate.

Line 439: A rapid onset of inexplicable asthenia and low sodium plasma levels should lead to rule out Hypophysitis occurrence.

From table 9 and 10, ≥ 1 Hypophysitis should be ruled out in the present 5 % of patients, with unfavorable characteristics   are alive at 60 months, it will be important, for clinical decisions, to know how many of them with PS3, associated to steroids and PPI use, are alive at 60 months.

Also, in the discussion I would suggest focusing on patients  usually excluded by CRT who derived a clinical benefit  from the treatment.

Author Response

The authors kindly appreciated and were deeply grateful for the insightful and thoroughly pertinent comments toward this manuscript. Thus, we had the opportunity to further improve our article and to provide the best version possible to the reading medical audience. The authors clearly corrected all the points indicated by the reviewer. We carefully tracked the changes we had made to be visible and easy to recognize. In order to send the appropriate conclusion from our research, we thank the reviewer for his/her hard work and dedication to good publishing interests. The reviewer is gently asked to check the following corrections, listed below.

The authors stated that RWD studies are necessary to generate high-quality RWE for subpopulations under-represented in RCTs, and I agree.
1. I am asking the authors to provide   the incidence of   endocrinopathies in the whole   population and in patients differing from those of RCTs.   For instance: WHO PS 3, pretreatment, brain met, active infections, concomitant tumors, steroid use > 10 mg daily of prednisone or equivalent, etc. In table 1 for instance the steroid use is not included. The authors thank the reviewer for this important point, previously missed. We have accordingly revised and added relevant information on page 17, paragraph 2, in lines 453-472. The steroid use is described with supplementary details in table 2, on page 7 and 8.

  1. Line 199: “Half of them were smokers and for 48.8% of the patients smoking was not toxic exposure” could you clarify please? Thank you for this clarification. Indeed, 48.8% of patients were not smokers and the previous sentence was vague and we deleted it.
  2. Lines 229-230    How cortisol and ACTH evaluation was performed in patients already on steroids? Thank you for this very pertinent question. Please find the explaining sentence on page 18, paragraph 1, in lines 500-502. In patients already on steroids for brain metastasis, immune-related PAI could not be suspected.
  3. Table 3: Transitory hyperthyroidism is usually a short window of subsequent hypothyroidism, sometimes captured, sometimes not, in my opinion it should not be calculated except when it persists, and that event is quite rare from literature and from my experience. Moreover, how did you include the transient modification of TFTs? Thank you for this important argument. TFTs are evaluated at baseline and before each ICI cycle of treatment and we followed guidelines protocols. We reported patients with transitional modified TFTs to better acknowledge the possibility of temporary effect of ICIs on thyroid function. Please find the entire relevant text on page 17, paragraph 3, between lines 477-485.
  4. Table 4: clarify better the patient’s number and the events number, otherwise it is not clear and immediate. Thank you for this clarification. 55 lung cancer patients experienced 65 irAEs, 47 patients experienced 51 endocrine-irAES and 13 patients had 14 other irAEs, as described on page 8, paragraph 1, in lines 253-259, 260-262 and lines 267-269.
  5. Table 5: treatment duration and treatment response indicate that patients without progression remained on therapy longer and obviously could develop more AES, in table 6 only response to ICI remained significant and this is confirmation. Moreover, the positive effect of surgery could be related to a clinical more favorable disease situation allowing these patients to be operated on and not to the surgery itself. Thank you for this clarification. We agree this comment. Indeed, the positive effect of surgery is not due to surgery itself, but to early-stage disease and more favorable prognostic factor. Please find the correction on page 19, first paragraph, between lines 547-553.
  6. Table 7: PDL1≥50% effect is probably related to a better response, can you confirm that? Thank you for pointing out this important fact. Indeed, PD-L1 status is closely correlated with ICIs efficacy and positive status increased survival. Please find this statement on page 12, paragraph 1, in line 324.
  7. In table 9: the significative effect of irAES on survival has been often reported, and related to multiple factors, among them the longer exposure of responding patients to treatment which increases the incidence of irAEs and the underlining common mechanisms of tumor response and irAES. Moreover, with endocrine irAES we do not have the detrimental effect of the steroids needed to counteract the toxicity. Thank you for pointing out this fascinating research topic. Please find the author’s argumentation on page 18, paragraph 2, in lines 525-539.
  8. 363: indicate 9LA regimen as combo immune (NIVO/IPI) Thank you for this important amendement. Please find the correction on page 15, paragraph 2, in lines 378-379.
  9. Lines 368-370: do you refer to all ECOG PS 3 patients of the table 1 or 23 % of them who received only one-two cycles? If all patients with PS3 received only 1-2 cycles, their exclusion from the eligibility is justified, in my opinion. Thank you for this opinion. Indeed, these patients should have been excluded from the statistical analysis. Taking into consideration the small sample of lung cancer patients and the reality of clinical practice in Romania, we decided to include them. Please find the argumentation on page 15, second paragraph 2, lines 384-385 and page 16 first paragraph, in lines 386-390.
  10. Line411: Hypophysitis is usually treated with substitutive therapy with Cortone Acetate. Thank you for this question. In Romania there is available for use only Prednisone, equivalent dose 5 mg Prednisone = 20 mg Hydrocortisone acetate. Please find the correction on page 9, in table 3, in line 277 and on page 18, first paragraph, in lines 505-508.
  11. Line 439: A rapid onset of inexplicable asthenia and low sodium plasma levels should lead to rule out Hypophysitis occurrence. Thank you for this point. Please find out the correction on page 17, third paragraph, lines 492-493 and page 18, first paragraph between lines 494-495.
  12. From table 9 and 10, ≥ 1 Hypophysitis should be ruled out in the present 5 % of patients, with unfavorable characteristics   are alive at 60 months, it will be important, for clinical decisions, to know how many of them with PS3, associated to steroids and PPI use, are alive at 60 months. Thank you for this observation. Taking into consideration that ECOG 3/PS 3 patients were treated with just 1 or 2 cycles of immunotherapy, no patient in this cohort were alive at 60 months. Please find the correction on page 12, first paragraph, between lines 324-326.
  13. Also, in the discussion I would suggest focusing on patients usually excluded by CRT who derived a clinical benefit from the treatment. Thank you for this relevant point, as our study is a real-life cohort study. Please find the discussion on page 19, second paragraph, between lines 554-579.

Reviewer 2 Report

Comments and Suggestions for Authors

Coniac et al provided a real-world evaluation of immune-related endocrine adverse events (irAEs) in metastatic non-small-cell lung cancer (mNSCLC) patients treated with ICI therapy.

Although all study population has mNSCLC, it is a diverse population with different histologic subtypes, different treatment regimens, PD-1 status, metastatic sites and numbers. Please discuss the limitations of this heterogeneous cohort more explicitly in the manuscript.

Table 2 shared some characteristics however the exact time of these data is unclear; are they the last timepoint characteristics?

It is important to distinguish diabetes mellitus and ICI-induced type 1 diabetes. Some patients may have type 2 diabetes at baseline (high c-peptide) and it should be confirmed that they did not have T1D at baseline.

It is unclear if the proportional hazard assumptions were tested for Cox regression models. No correction/adjustment for multiple comparisons is mentioned.

The impact of steroids and PPIs on ICI response is noted, but causality cannot be concluded without more rigorous statistical adjustments.

The study does not appear to stratify patients by the reason for PPI or steroid use and suggests that PPIs and steroids negatively impact ICI efficacy, but this is based on association rather than causation. Patients receiving PPIs or steroids may have had worse baseline conditions (e.g., brain metastases requiring steroids), making it unclear if these medications cause reduced survival or if they are markers of more severe disease.

Minor points:

The results contain long descriptive paragraphs that reduce readability. Referring to tables and summarizing some parts of tables can improve readability.

There are 2 patient discrepancy between figure one and the text. 489 vs 487 and 239 vs 237.

Author Response

The authors kindly appreciated and were deeply grateful for the insightful and thoroughly pertinent comments toward this manuscript. Thus, we had the opportunity to further improve our article and to provide the best version possible to the reading medical audience. The authors clearly corrected all the points indicated by the reviewer. We carefully tracked the changes we had made to be visible and easy to recognize. In order to send the appropriate conclusion from our research, we thank the reviewer for his/her hard work and dedication to good publishing interests. The reviewer is gently asked to check the following corrections, listed below.

Coniac et al provided a real-world evaluation of immune-related endocrine adverse events (irAEs) in metastatic non-small-cell lung cancer (mNSCLC) patients treated with ICI therapy.

  1. Although all study population has mNSCLC, it is a diverse population with different histologic subtypes, different treatment regimens, PD-1 status, metastatic sites and numbers. Please discuss the limitations of this heterogeneous cohort more explicitly in the manuscript. Thank you for this relevant point, as our study is a real-life cohort study. Please find the discussion on page 19, second paragraph, between lines 554-579.
  2. Table 2 shared some characteristics however the exact time of these data is unclear; are they the last timepoint characteristics? Thank for your clarifying question. Table 1 presents baseline characteristics before ICI initiation. Please find specific text on page 4, second paragraph, in lines 157-163. Table 2 presents patients characteristics during ICI treatment, such as treated infections, steroids, PPI and opioids usage. Please find the clarifying text on page 7, first paragraph, in lines 214-216. BMI, dyslipidemia and DM were assessed at baseline.
  3. It is important to distinguish diabetes mellitus and ICI-induced type 1 diabetes. Some patients may have type 2 diabetes at baseline (high c-peptide) and it should be confirmed that they did not have T1D at baseline. Thank you for this crucial point. Please find the explanation on page 18, first paragraph, between lines 508-515.
  4. It is unclear if the proportional hazard assumptions were tested for Cox regression models. No correction/adjustment for multiple comparisons is mentioned.

We thank the reviewer for such careful scrutiny of the statistical methods that we employed, as respecting assumptions is often overlooked in medical research. In our opinion, this is of utmost importance for the relevance of the results presented. 

The proportional hazard assumption was tested for the Cox regression models. We excluded predictors where Kaplan-Meier plots crossed, as this is a mark of violating the proportional hazards assumption. We calculated Schoenfeld residuals using the cox.zph() function in R, checking both the p-value of the function and the plot of residuals1. For brevity, we only reported the predictors that met these assumptions.

We did not adjust for multiple comparisons as our research is exploratory in nature, it investigates if the parameters we collected influence clinical outcomes. Therefore, such corrections are not appropriate for our design. As stated by Chen et al., “Type I errors and type II errors are negatively correlated”. Using the Bonferroni method or other adjustment methods would decrease the probability of type I error while increasing the probability of type II errors. In our case, we were interested in controlling the probability of having type II errors, as we intend to generate hypotheses for further investigation.

Thank you for mentioning this issue; it improves the transparency of our study. We added an explanation on why we chose not to adjust for multiple comparisons on page 3, paragraph 8, in lines 129-130.

  1. Dessai S, Patil V. Testing and interpreting assumptions of COX regression analysis. Cancer Research, Statistics, and Treatment. 2019;2(1):108-111. doi:10.4103/CRST.CRST_40_19
  2. Althouse AD. Adjust for Multiple Comparisons? It’s Not That Simple. Annals of Thoracic Surgery. 2016;101(5):1644-1645. doi:10.1016/j.athoracsur.2015.11.024
  3. Chen SY, Feng Z, Yi X. A general introduction to adjustment for multiple comparisons. J Thorac Dis. 2017;9(6):1725-1729. doi:10.21037/jtd.2017.05.34

  1. The impact of steroids and PPIs on ICI response is noted, but causality cannot be concluded without more rigorous statistical adjustments. Thank you for this clarifying point. Indeed, the authors do not suggest in conclusions that there is a causality between steroids and PPI usage and ICI’s response. We just showed a negative impact on ICI’s efficacy. Please find the clarifying text on page 16, second paragraph, between lines 421-439 and on page 17, first paragraph between lines 440-442.
  2. The study does not appear to stratify patients by the reason for PPI or steroid use and suggests that PPIs and steroids negatively impact ICI efficacy, but this is based on association rather than causation. Patients receiving PPIs or steroids may have had worse baseline conditions (e.g., brain metastases requiring steroids), making it unclear if these medications cause reduced survival or if they are markers of more severe disease. Thank you for this important clarifying subject. Please find the explanation text on page 16, second paragraph, between lines 427-439 and on page 17, first paragraph, between line 440-442.

Minor points:

  1. The results contain long descriptive paragraphs that reduce readability. Referring to tables and summarizing some parts of tables can improve readability. Thank you for this important point. The authors compressed as much as possible the descriptive results paragraphs. Taking into consideration the multiple parameters included in three tables, we described information not written in Tables 2, 3 and 4, such as type of secondary primary tumors, or concomitant infections or type of opioids.
  2. There are 2 patient discrepancy between figure one and the text. 489 vs 487 and 239 vs 237. Thank you for identifying this minor mistake. We corrected the figure.

Round 2

Reviewer 1 Report

Comments and Suggestions for Authors

My question points have been addressed and answered. I still think that prednisone is not adequate as substitutive therapy for Hypophysitis, from a clinical point of view and for the patient’s wellbeing, and clinicians treating patients with ICIs should find a way to provide Cortone acetate for the patients needing it.

Another point I would like to stress again is the difference between endocrinopathies and all others Immune related toxicities. In other toxicities, the steroids or immunosuppressants need could have detrimental effects on the antitumor efficacy, and this effect could be dose and time-exposure related. While, in presence of endocrinopathies, the substitutive therapy is not expected to counteract or impact the antitumor activity of ICI.

Anyway, I acknowledge the retrospective nature of the study and the efforts put in collecting and submitting these RWD, and I approve this updated version

Author Response

Revision 2 – Cancers-3480207 – 28.03.2025 - SC

Reviewer 1

  1. My question points have been addressed and answered. I still think that prednisone is not adequate as substitutive therapy for Hypophysitis, from a clinical point of view and for the patient’s wellbeing, and clinicians treating patients with ICIs should find a way to provide Cortone acetate for the patients needing it.

    Thank you for this important treatment aspect of patients with endocrine-irAEs. You are absolutely right and we corrected the text in page 19, paragraph 2, between lines 442 – 446.

  2. Another point I would like to stress again is the difference between endocrinopathies and all others Immune related toxicities. In other toxicities, the steroids or immunosuppressants need could have detrimental effects on the antitumor efficacy, and this effect could be dose and time-exposure related. While, in presence of endocrinopathies, the substitutive therapy is not expected to counteract or impact the antitumor activity of ICI.

    Indeed, this specification is crucial for better understanding of the difference between Endocrine-irAEs and all other irAEs. We agreed upon this correction and we are grateful for pointing out this important aspect. Please find out the rectification in page 19, paragraph 2, between lines 442-446.

  3. Anyway, I acknowledge the retrospective nature of the study and the efforts put in collecting and submitting these RWD, and I approve this updated version.

We gratefully appreciate the reviewer for the entire effort and support to improve our article and we kindly acknowledge all we have learnt from this extensive review. In our future research we will keep in mind to follow clear but detailed points, as provided in this review.

Reviewer 2 Report

Comments and Suggestions for Authors

Authors addressed all my previous comments. The flow of the manuscript still needs to be improved.

In the results, please indicate Table 1 in the “Patient demographic and clinical characteristics assessed at baseline (meaning the initiation of ICI’s therapy) included age, sex, body mass index (BMI) dyslipidemia and Diabetes Mellitus, Eastern Cooperative Oncology Group (ECOG) performance status, histological features, number and sites of metastasis at baseline, presence of EGFR mutations or ALK rearrangements, PD-L1 status (TABLE 1)” sentence. Treatment characteristics are not part of baseline characteristics. Please add type of ICI used in first and second line, number of ICI’s cycles, response to ICIs, type of therapy before ICI, radiotherapy for metastatic sites to the treatment table (table 2). BMI, smoker status, DM dyslipidemia are baseline characteristics, please move them to table 1. By doing these, you can truly separate baseline (before treatment) and after/during treatment phases in two tables.

In the discussion you mentioned some limitations before “Our research has several limitations to be listed.” sentence which disrupts the flow.

Author Response

Revision 2 – Cancers-3480207 – 28.03.2025 - SC

Reviewer 2

  1. Authors addressed all my previous comments. The flow of the manuscript still needs to be improved.

    We gratefully appreciate the reviewer for the entire effort and support to improve our article and we kindly acknowledge all we have learnt from this extensive review. In our future research we will keep in mind to follow clear but detailed points, in a logic and coherent flow, as provided in this review. We revised the text accordingly.
  2. In the results, please indicate Table 1 in the “Patient demographic and clinical characteristics assessed at baseline (meaning the initiation of ICI’s therapy) included age, sex, body mass index (BMI) dyslipidemia and Diabetes Mellitus, Eastern Cooperative Oncology Group (ECOG) performance status, histological features, number and sites of metastasis at baseline, presence of EGFR mutations or ALK rearrangements, PD-L1 status (TABLE 1)” sentence. Treatment characteristics are not part of baseline characteristics. Please add type of ICI used in first and second line, number of ICI’s cycles, response to ICIs, type of therapy before ICI, radiotherapy for metastatic sites to the treatment table (table 2). BMI, smoker status, DM dyslipidemia are baseline characteristics, please move them to table 1. By doing these, you can truly separate baseline (before treatment) and after/during treatment phases in two tables.

    Indeed, the information should have been more clearly presented and we agreed on this issue. Thank you for pointing out this aspect, as previous tables might have been confusing for the reading audience. We have corrected the information and split all data in 3 separate tables. Please find Table 1 (baseline characteristics) on page 5 and 6, Table 2 (treatment characteristics) on page 6 and 7, Table 3 (clinical features during immunotherapy) on page 9 and 10.
  3. In the discussion you mentioned some limitations before “Our research has several limitations to be listed.” sentence which disrupts the flow.

    Thank you for this point. We have accordingly corrected the text. Please find the correction on page 22, paragraph 3, between lines 591-602.
